# Modification Strategies for Ionic Complementary Self-Assembling Peptides: Taking RADA16-I as an Example

**DOI:** 10.3390/polym14235221

**Published:** 2022-11-30

**Authors:** Weiwei Guo, Yinping Ma, Lei Hu, Yujie Feng, Yanmiao Liu, Xuedong Yi, Wenzhi Zhang, Fushan Tang

**Affiliations:** 1Department of Clinical Pharmacy, Key Laboratory of Basic Pharmacology of Guizhou Province and School of Pharmacy, Zunyi Medical University, Zunyi 563006, China; 2Key Laboratory of Basic Pharmacology of Ministry of Education and Joint International Research Laboratory of Ethnomedicine of Ministry of Education, Zunyi Medical University, Zunyi 563006, China; 3The Key Laboratory of Clinical Pharmacy of Zuni City, Zunyi Medical University, Zunyi 563006, China; 4School of Preclinical Medicine, Zunyi Medical University, Zunyi 563006, China; 5Department of Pharmacy, Affiliated Hospital of Zunyi Medical University, Zunyi 563000, China

**Keywords:** hydrogel, active motifs modification, regenerative medicine, self-assembling peptide

## Abstract

Ion-complementary self-assembling peptides have been studied in many fields for their distinct advantages, mainly due to their self-assembly properties. However, their shortcomings, such as insufficient specific activity and poor mechanical properties, also limited their application. For the better and wider application of these promising biomaterials, ion-complementary self-assembling peptides can be modified with their self-assembly properties not being destroyed to the greatest extent. The modification strategies were reviewed by taking RADA16-I as an example. For insufficient specific activity, RADA16-I can be structurally modified with active motifs derived from the active domain of the extracellular matrix or other related active factors. For weak mechanical properties, materials with strong mechanical properties or that can undergo chemical crosslinking were used to mix with RADA16-I to enhance the mechanical properties of RADA16-I. To improve the performance of RADA16-I as drug carriers, appropriate adjustment of the RADA16-I sequence and/or modification of the RADA16-I-related delivery system with polymer materials or specific molecules can be considered to achieve sustained and controlled release of specific drugs or active factors. The modification strategies reviewed in this paper may provide some references for further basic research and clinical application of ion-complementary self-assembling peptides and their derivatives.

## 1. Introduction

Hydrogel is formed by cross-linking of polymer materials in water [1]. Peptide-based materials are a distinctive type of hydrogel material. For many years, peptide-based hydrogels have received a great deal of attention from researchers in various fields due to their good biocompatibility, high water content, and excellent injectability, and modification of the related peptides has been considered an important and interesting task for the development of peptide-based hydrogels [2,3]. Based on the structural characteristics of peptides, different structural and functional properties of peptide-based hydrogels can be obtained by modifying the types or sequences of the peptides with different amino acids [4,5]. From this concept, different types of peptide-based hydrogels, such as the short peptide, ultrashort peptide, multicomponent matrices, and all-aromatic peptide hydrogels, have been developed profoundly [6,7,8].

Ion-complementary self-assembling peptides are composed of alternately arranged charged hydrophilic and hydrophobic amino acids and can self-assemble into supramolecular nanofibrous hydrogels [9]. Many ion-complementary self-assembling peptides with superior performance have been designed as promising biomaterials after EAK16-II [10], such as RADA16-I [11], KLD12 [12], and KFE8 [13] (Table 1). Among these, RADA16-I is the most studied typical ion-complementary self-assembling peptide in various fields, such as tissue engineering [14,15,16], drug delivery [17,18], three-dimensional cell culture [19,20,21,22], and hemostatic applications [23].

Compared with other hydrogel materials, in addition to good biocompatibility, low immunogenicity, biodegradability, and non-toxic degradation products [24], ion-complementary self-assembling peptides represented by RADA16-I have several special advantages as a promising biomaterial: (1) they can self-assemble into hydrogels by simply adjusting pH or adding salt, without adding various cross-linking agents that may have harmful effects on the body [25], (2) the self-assembling hydrogel structure has a suitable pore size, which can simulate the three-dimensional microenvironment of the extracellular matrix, support the attachment of various mammalian cells, and provide a suitable microenvironment for cell proliferation and migration [26], (3) the two ends of the peptide chain can be modified by various active short peptide motifs by solid-phase synthesis and the process is relatively easy to control and characterize [27] (Figure 1).

Although RADA16-I has many unique advantages, its shortcomings cannot be ignored. RADA16-I lacks specific tissue biological activity, making it difficult to be applied to various tissues or disease environments in a targeted manner [28]. Its hydrogel structure is maintained by a variety of weak interaction forces (hydrophilic/hydrophobic interaction, hydrogen bonding, van der Waals forces, etc.), resulting in poor mechanical properties [29]. Although RADA16-I has been shown to have a certain sustained-release effect for the delivery of hydrophobic drugs, there is still improvement room for the delivery of other types of drugs [30]. Modifications specific to these shortcomings should be researched and implemented to make RADA16-I—represented ion-complementary self-assembling peptides more suitable for practical applications.

There have been numerous and complicated studies on ion-complementary self-assembling peptides in related fields. RADA16-I covers most of the research areas of ionic complementary self-assembling peptides and is representative of this type of peptide. Many articles about the modification of RADA16-I structure also allow us to sort out the generic modification strategies of this class of hydrogel-forming peptide by taking RADA16-I as a clue (Table 2). We, therefore, took RADA16-I as an example to summarize the modification of the structure of ionic complementary self-assembling peptides (Figure 2). We aimed to display modification strategies for ionic complementary self-assembling peptides by reviewing research on the modification of RADA16-I and to provide a useful reference for further related research and application.

**Figure 1 polymers-14-05221-f001:**
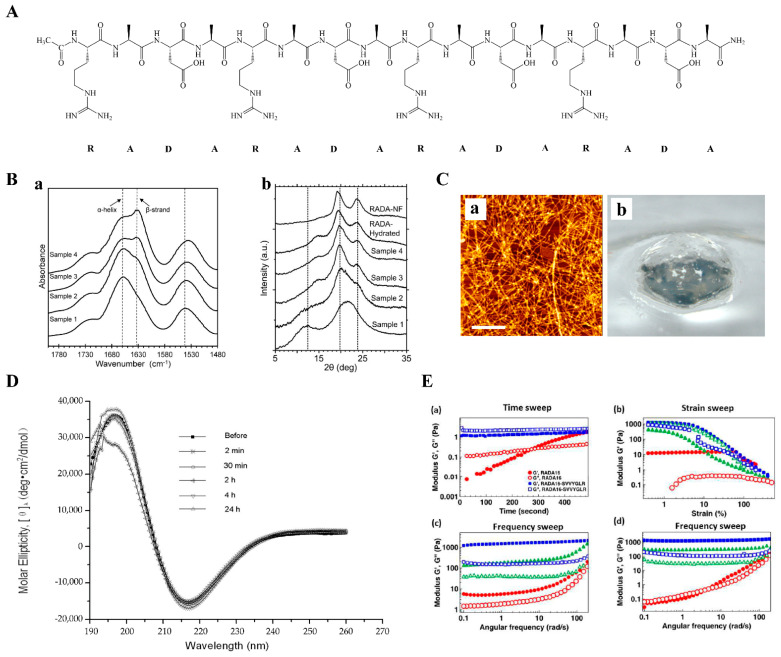
The general properties of RADA16-I. (**A**) Chemical structures of RADA16-I. (**B**) FTIR spectra for RADA16-I at different temperatures (**a**). WAXD diffractograms for Samples 1–4, RADA-Hydrated, and RADA-NF (**b**). The temperatures from Samples 1–4 are 298 K, 344 K, 347 K, and 363 K, respectively. ‘RADA-Hydrated’ refers to Sample 1, hydrated by adding 1 mg H_2_O/1 mg of the peptide. ‘RADA-NF’ refers to the sample that has been self-assembled in the aqueous salt solution. Adapted with permission from Ref. [31]. Copyright 2013 Journal of Peptide Science. (**C**) Atomic force microscope (AFM) image (**a**) and macroscopic image (**b**) of RADA16. (**D**) CD examination of the peptide structures at various times before and after sonication: 2 min, 30 min, 2 h, 4 h, and 24 h. Adapted with permission from Ref. [32]. Copyright 2012 International Journal of Molecular Sciences. (**E**) Rheological behavior of RADA16-I and RADA16-SVVYGLR (a functionalized sequence of RADA16). (**a**) Time sweeps of 1% (*w*/*v*) RADA16-I and RADA16-SVVYGLR. (**b**) Strain sweeps of 1% (*w*/*v*) RADA16 and RADA16-SVVYGLR in various pH conditions. ●: RADA16-I in pH3; ▲: RADA16-I in pH7; ◼: RADA16-I in pH13; ○: RADA16-SVVYGLR in pH3; △: RADA16-SVVYGLR in pH7; □: RADA16-SVVYGLR in pH13. Frequency sweep of 1% (*w*/*v*) RADA16-I (**c**) and RADA16-SVVYGLR (**d**) in various pH conditions. Symbols for storage modulus (G’): ●: pH3; ▲: pH7; ◼: pH13; for loss modulus (G”): ○: pH3; △: pH7; □: pH13. (*n* = 3). Adapted with permission from Ref. [33]. Copyright 2017 Nanoscale.

## 2. Active Modification of RADA16-I: Modification with Active Motifs

Sloid-phase synthesis can directly modify both ends of the RADA16-I peptide chain with active motifs. This modification method is relatively easy to operate and characterize, and a large number of studies have attempted to modify RADA16-I with various biologically active motifs to form functionalized RADA16-I to enhance the biological activity of RADA16-I. The active motifs derived from the extracellular matrix were introduced to simulate the functions of the extracellular matrix (such as cell adhesion). The motifs from other sources can also be introduced to indirectly promote or maintain cell function by resisting the harmful effects of the materials on cells and tissues.

Modifying the peptide structure with these active motifs may affect the intrinsic properties of RADA16-I. Minimizing the adverse effect of active motifs on the self-assembly properties of RADA16-I is a challenging issue that must be considered before designing functionalized RADA16-I.

### 2.1. Influencing Factors of Active Motifs on Self-Assembling of RADA16-I

The self-assembling performance of RADA16-I can be influenced by many factors, such as the number of amino acid residues, hydrophobicity, net charge, hydrogen bonding, and van der Waals forces of the introduced reactive motifs. Among them, the hydrophobicity of the active motif, the charge, and the length of the motif are relatively controllable. These factors may not only affect the self-assembling of RADA16-I but also determine whether the biological activity of the modified active motifs can be retained.

#### 2.1.1. Hydrophilic and Hydrophobic Property

The best motifs should have an alternating arrangement of hydrophobic and hydrophilic units to achieve a moderate hydrophilic-hydrophobic balance. Modified sequences with too hydrophilic or too hydrophobic active motifs will be difficult to completely self-assemble. The self-assembling properties of three RADA16-I functionalized sequences, RADA16-ALK, RADA16-SDE, and RADA16-BMHP1, were compared in a study, and the results showed that the overly hydrophobic RADA16-ALK underwent rapid hydrophobic collapse to form spherical aggregates rather than long fibers, while the highly hydrophilic SDE motif added to RADA16-I perturbed the otherwise stable structure of RADA16-I and RADA16-I-SDE did not form organized nanostructures, and only RADA16-BMHP1 with moderate hydrophilicity and hydrophobicity maintained its typical β-sheet structure and self-assembled into nanofibers. The differences were believed to be caused by the different hydrophilicity of the motifs [34]. It was also indicated that overly hydrophobic motifs were less active after modification due to their inability to be exposed to the environment [35].

#### 2.1.2. Charge

Active motifs with a net charge can lead to discrete and length changes in functionalized RADA16-I fibers due to charge repulsion. Simple additions of lysine and aspartic acid were used to design RADA16-I with a range of net charges (two, four, and six positive or negative charges, respectively), and all functionalized RADA16-I were observed to self-assemble into nanofibers under AFM [36]. The peptide sequences with two positive charges formed numerous but short fibers and aggregated into bundles. As the positive net charge increases, the fibers become longer, progressively looser, and separate into single fibers. A similar situation was observed for the negatively charged sequences, with the difference that as the net negative charge increased, the fibers became progressively shorter but did not completely separate into single fibers. This indicates that the charged nature and the number of active motifs can exert an important influence on the self-assembling of RADA16-I.

#### 2.1.3. Length of the Main Chain of the Active Motif

Theoretically, the fewer amino acid residues in the active motif, the better self-assembling performance the motif-modified functionalized RADA16-I will have. The number of specific residues in the modification motif that does not disrupt the self-assembling of short peptides needs to be discussed in context. It is shown that adding no more than 12 additional residues on the self-assembling peptide RADA16-I did not inhibit the self-assembling properties and the formation of nanofibers [37]. Many studies have also borrowed this idea when designing modification motifs. However, the relevant research data also showed that only a few motifs, such as RGD, RGDSP, YIGSR, and SVVYGLR, can self-assemble independently, and most of the other motif-modified peptides had a poor self-assembling performance. It is often necessary to promote their self-assembly by some complementary measures.

### 2.2. Countermeasures for Affecting Self-Assembling Properties Caused by Motif Introduction

Even after knowing the controllable factors of the effect of active motifs on RADA16-I self-assembling, researchers tend to pay more attention to the activity changes brought about by the motifs. Some countermeasures, including but not limited to introducing glycine, mixing the modified peptide with the original RADA16-I, and extending the chain length of RADA16-I, may be required to maintain the intrinsic self-assembling properties of RADA16-I.

#### 2.2.1. Adding Glycine

As the simplest amino acid, glycine is often used as a spacer motif to connect RADA16-I with the active motif. The addition of the glycine spacer motif can increase the freedom of the active motif, leading to the expansion of the β-sheet structure between peptide molecules, thus contributing to the stability of the overall supramolecular structure. The exposed active motif can also facilitate the binding of modified peptides to the receptor. To investigate the effect of the number of glycine spacer motifs (Gs) on the self-assembling properties and biological activity of functionalized RADA16-I, Francesca Taraballi et al. designed three sequences RADA16-I-PFSSTKT (0G-BMHP1, without Gs), RADA16-I-GGPFSSKTK (2G- BMHP1, 2 Gs) and RADA16-I-GGGGPFSSTKT (4G-BMHP1, 4 Gs) by adding different amounts of glycine between RADA16-I and the active motif PFSSKTK. An atomic force microscope observed that nanofibers formed by functionalized peptides with two and four glycines as spacer groups were longer than those formed by functionalized peptides without glycine. The Fourier Transform Infrared (FTIR) results showed that the functionalized peptide containing four glycine spacer motifs showed the most intermolecular β-Sheet structure, while the peptide lacking the glycine spacer was a loose assembly. Thermal stability experiments showed that at approximately 100 °C, 2G-BMHP1 and 4G-BMHP1 lost 28% of the initial β-sheet structure, while 0G-BMHP1 lost 34% of the β-sheet structure. This indicated that the addition of glycine spacer motifs resulted in better thermal stability of the functional peptide. Subsequent cell activity assays also showed that 4G-BMHP1 was more effective in stimulating the adhesion of neural stem cells and improved their survival in differentiation media. Combined with the Raman results, it can be easily analyzed that the bioactive motifs are more significantly exposed to the solvent due to the prolongation of the glycine spacer group, thus exerting better bioactivity [38].

#### 2.2.2. Mixing with RADA16-I

Mixing functionalized RADA16-I with unmodified RADA16-I in a certain ratio can maintain the self-assembling of functionalized RADA16-I. Although there is a lack of research on the detailed principle of this measure, this method is undoubtedly the most applied, straightforward, and effective remedy when self-assembling is destroyed. Most related studies used this method when the self-assembly of RADA16-I was disrupted after introducing active motifs. Lei Lu et al. designed a functionalized sequence (RADA)4-GG-(PAMP-12) and mixed it with pure RADA16-I in different proportions of the functionalized sequence (0%, 2.5%, 5%, 10%, 20%, 40%, 80%, 100% *w*/*w*). The AFM results showed that the nanofiber network with the addition of 10% (RADA)4-GG-(PAMP- 12) was generally able to maintain a similar structure to that of pure RADA16-I. However, when the ratio of (RADA)4-GG-(PAMP-12) was further increased, the length of nanofibers became shorter, resulting in less entanglement of nanofibers. 100% (RADA)4-GG-(PAMP-12) was not even able to form effective nanofibers but only nanoparticles. The height of nanofibers formed with 100% (RADA)4-GG-(PAMP-12) was 2.5–3.5 nm, which was much higher than the height of nanofibers with pure RADA16-I (∼1.5 nm). This was reflected in the macroscopic morphology, where 20% (RADA)4-GG-(PAMP-12) was able to form stable hydrogels, and containing 40% made it a weak hydrogel, while 80% and 100% (RADA)4-GG-(PAMP-12) can only exist in a liquid state [39]. The functionalized peptide RAD-RGI was designed by adding the bioactive motif RGIDKRHWNSQ to the C-terminus of RADA16-I. Circular dichroism (CD) measurements analysis revealed that the secondary structure of the functionalized peptide RAD-RGI was disrupted by the added active motif. The β-sheet content of the pure functionalized peptide RAD-RGI solution was only 23.5% relative to the β-sheet content of pure RADA16. However, when an equal proportion of pure RADA16 was added, the β-sheet content increased to 47.1%. By AFM, only small nanoparticles or nanorods, but not nanofibers, were observed in the pure RAD-RGI solution. Long, uniform, and interwoven nanofibers were formed when an equal proportion of RADA16-I was added [40]. From the current study, the ratio of mixing was mostly 1:1; when the self-assembling performance of functionalized RADA16-I was unsatisfactory, the ratio of RADA16-I was increased to maintain the self-assembling performance of functionalized RADA16, as one study increased this ratio to 4:1 [41].

#### 2.2.3. Prolonging the Original Sequence

The self-assembling properties of functionalized RADA16-I can be maintained by extending the sequence. However, only one study [42] employed this measure. The researchers of this study extended the sequence of RADA16-I to RADA32 and then covalently bound the bioactive sequence GIGAVLKVLTTGLPALISWIKRKRQQ at the C-terminus of RADA32 to form the functionalized sequence MR and loaded with doxorubicin (MRD). CD measurements showed that the maximum molar residual ellipticity at 216 nm was negative for both MR and MRD, indicating the formation of β-sheet structures. MR and MRD can form interwoven nanofiber networks with diameters of 10.5 ± 1.8 and 21.2 ± 4.2 nm, respectively, as observed by transmission electron microscopy (TEM) images. The rheological analysis also showed that these nanofibers could further form stable hydrogels with good reassembling properties and ease of injection application. Although it also achieved good results, this method is undoubtedly more cumbersome and expensive than the two measures mentioned above.

### 2.3. Modification with Active Motifs

#### 2.3.1. Motifs from the Extracellular Matrix (ECM)

The extracellular matrix can generally be divided into three categories: proteoglycans, structural proteins (collagen and elastin), and adhesive proteins (fibronectin and laminin). Proteoglycan can form a jelly and fill in the extracellular matrix so that the cells have a strong anti-extrusion ability; structural proteins are the skeleton of the extracellular matrix, providing tension and elasticity to the cells, and adhesion proteins can mediate cell adhesion and stimulate cell movement migration and differentiation. Modification of RADA16-I with the active domain of the extracellular matrix can directly promote the development of cells in beneficial directions (adhesion, migration, etc.) [43,44,45,46].

Promoting the adhesion of materials to cells is the main direction for improving or modifying biological materials. Among the three types of extracellular matrices, most of the sequences currently used for material modification are derived from the adhesion proteins that mediate cell adhesion: fibronectin and laminin. RGD was originally identified as a sequence on fibronectin that promotes cell adhesion by binding to integrin receptors on the cell surface, but this sequence was later found to be present in other adhesion proteins [47]. Due to its cell-adhesive activity and ultra-short sequence of only three amino acids, RGD, as a modified sequence of materials, is generally considered one of the most widely and frequently used classical sequences [48,49]. The ring-shaped RGD, cRGD, was shown to have a stronger binding affinity to neural axons and was more stable. Compared with unmodified RADA16-I, RADA16-I containing the C (RGDFK) modification motif exhibited better neural adhesion and promoted the cell proliferation of neural precursor cells [50].

Another RGD mimetic peptide, PRGDSGYRGDSG (PRG), derived from collagen type VI, may exhibit stronger cell adhesion because it contains two repetitive RGD sequences. Researchers compared the activity of PRG-functionalized RADA16-I (RADA16-PRG) with that of unmodified RADA16-I and found that RADA16-PRG not only promoted the adhesion and proliferation of pre-osteoblast MC3T3-E1 cells but also increased alkaline phosphatase (ALP) activity, an early marker of osteoblast differentiation. More importantly, MC3T3-E1 cells cultured on RADA16-PRG scaffolds had a higher osteocalcin concentration compared with other functionalized RADA16-I after 14 days [51]. Rat periodontal ligament-derived cells were cultured in RADA16-PRG hydrogels to investigate the potential of RADA16-PRG for the treatment of periodontal defects. After 72 h, cells in RADA16-PRG hydrogels showed significantly higher viability and proliferation than those in unmodified RADA16-I and other functionalized PDS hydrogels (Figure 3A,B). Subsequent in vivo studies demonstrated that topical application of RADA16-PRG hydrogels in a rat periodontal defect model increased bone volume fraction and promoted the healing of periodontal defects [52].

In addition to fibronectin, another extracellular matrix that mediates the adhesion of cells to materials: laminin, has also been derived with a number of sequences for modification of RADA16-I. IKVAV is a functional motif of the laminin molecule. SIKVAV-functionalized RADA16-I (R-GSIK) were used as hydrogel scaffolds for the transplantation of mesenchymal stem cells (MSCs) to investigate the effect of the combination on functional recovery and neuroinflammatory responses after traumatic brain injury (TBI) in rats. Behavioral scoring found that functionalized scaffolds loaded with MSCs enhanced TBI recovery in rats and significantly reduced the numbers of astrocytes and microglia, which release inflammation-related factors after TBI (Figure 3C). Subsequent Western blot analysis also found a reduction in related pro-inflammatory factors, confirming that the combination could improve stem cell survival after transplantation by inhibiting the inflammatory response [53].

Also derived from laminin, YIGSR-functionalized RADA16-I (RADA16-YIGSR) hydrogels have also been applied to transplant neural stem cells to study the therapeutic effect on Alzheimer’s disease (AD). Both in vitro and in vivo studies confirmed that the RADA16-YIGSR hydrogel scaffold promoted the survival and differentiation of Neural stem cells (NSCs), increased the secretion of anti-inflammatory and neurotrophic factors, and decreased the expression of pro-inflammatory factors (Figure 3D) [54]. A similar sequence SDPGYIGSR-functionalized RADA16-I was also used to transplant rat periodontal ligament-derived cells to investigate the effect on rat periodontal defect [52].

Osteopontin is widely present in the extracellular matrix. The angiogenic osteopontin-derived sequence SVVYGLR was covalently bound to RADA16-I (RADA16-SVVYGLR) to investigate the repair effect of this hydrogel scaffold on brain-damaged zebrafish. In vitro results of co-incubation of the hydrogels with neural cells (NSCs) showed that the RADA16-SVVYGLR hydrogels supported NSC survival and proliferation, and in vivo results demonstrated that the hydrogels promoted nerve and angiogenesis, helping damaged nerve tissue rebuild and promote brain function recovery [33]. Another osteopontin-derived sequence, DGRGDSVAYG bound to RADA16-I (RADA16- DGRGDSVAYG), has been used in bone regeneration with promising results [51].

However, not all functionalized RADA16-I achieved good results. Numerous active motifs were covalently bound to RADA16-I, and the result showed that some functionalized RADA16-I showed no activity [55]. It is believed that the biological function of functionalized scaffolds depends not only on the biological activity of functional motifs but also on the presentation forms of functional motifs, such as mobility, conformation, and orientation. The functionalized RADA16-I that showed no activity had an excess of hydrophobic amino acid residues, making it possible for the active motif to be extruded into the self-assembling nanofibers backbone and thus not enough active motifs being exposed to the environment to interact with cells.

**Figure 3 polymers-14-05221-f003:**
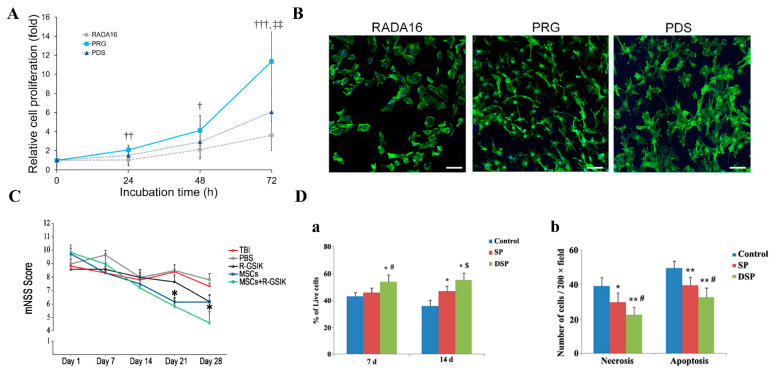
Motifs from the extracellular matrix. (**A**) Viability/proliferation of rat periodontal ligament-derived cells (rPDLCs). rPDLCs were cultured on the RADA16, PRG (RADA16-PRG), or PDS with medium or medium alone (control). WST-1 assay was used to assess cell viability/proliferation at indicated time points. In each group, values relative to those at 0 h were shown. Data presented as mean ± SD (*n* = 7). ^†^ *p* < 0.05, ^††^ *p* < 0.01, ^†††^ *p* < 0.001 compared with the RADA16 group, and ^‡‡^ *p* < 0.01 compared with the PDS group, by ANOVA with Tukey’s post hoc test. (**B**) Representative confocal laser scanning microscopy images. At 72 h, cells on PRG or PDS appear elongated and show cell protrusions. (Green; actin cytoskeleton). (Original magnification ×100, bar = 100 µm). Adapted with permission from Ref. [52]. Copyright 2021 Journal of Periodontal Research. (**C**) Modified neurological severity scores (mNSS) were performed in different groups. mNSS significantly improved in rats treated with MSCs+ R-GSIK and MSCs compared with TBI and PBS groups on day 21 after TBI. Adapted with permission from Ref. [53]. Copyright 2020 Cell and Tissue Research. (**D**) RADA16-YIGSR (DSP) increased the cell viability and protected NSC against the cytotoxicity of aggregated Aβ. (**a**) Cell viability was detected at 7 and 14 days after being seeded in DSP. NSCs in modified peptides showed increased cell survival compared to the unmodified RADA16 (SP). * *p* < 0.01 versus control; ^#^ *p* < 0.01 and ^$^ *p* < 0.05 versus SP. (**b**) Cell survival after Aβ1–40 treatment. * *p* < 0.05 and ** *p* < 0.01 versus control; ^#^ *p* < 0.05 versus SP. Adapted with permission from Ref. [54]. Copyright 2016 Molecular Neurobiology.

#### 2.3.2. Motifs from Other Sources

Apart from extracellular matrix-derived motifs, motifs from other sources are numerous and difficult to categorize. In order to present these studies better, we roughly divided these studies into two categories according to the main modification strategies for cell growth and disease treatment: promoting positive factors and resisting negative factors.

##### Promoting Positive Factors

The active motifs are mostly derived from the active structural domains of various proteins already present in the human body in this strategy. RADA16-I was modified with active motifs that directly enhance cell adhesion, proliferation, migration, and differentiation and promote cells to secrete secretions useful for disease treatment. Under this strategy, functionalized RADA16-I directly promotes the development of cells in a beneficial direction by enhancing their existing functions. It should be noted that the modification strategies mentioned in the “Extracellular matrix” section also belong to this category.

Angiogenesis is the basis for other functional reconstructions and repairs in tissue engineering and regenerative medicine. The regeneration of many tissues requires establishing a good vascular network as the supply of nutrients and oxygen [56]. The active motif KLT (KLTWQELYQLKYKGI) is a vascular endothelial growth factor (VEGF) mimetic peptide that acts as a VEGF agonist to activate VEGF-dependent signaling pathways. The effect of RAD/KLT (KLT-modified RADA16-I) hydrogels on human umbilical vein endothelial cells (HUVECs) was studied in vitro. The results showed that RAD/KLT significantly enhanced endothelial cell survival, adhesion, and migration. Subsequent chick chorioallantoic membrane (CAM) experiments also found that RAD/KLT exhibited better in vivo angiogenic activity in the absence of VEGF, inducing CAM tissue invasion and new capillary formation (Figure 4A) [57]. A similar result was obtained in another earlier related study [58].

Nerve regeneration is as important and fundamental as angiogenesis in tissue engineering and regenerative medicine. Unmodified RADA16-I has already shown great effects insupporting nerve regeneration, which provides a foundation for the application of functionalized RADA16-I in nerve regeneration [59]. RADA16-I was modified with the neurotrophic peptide RGI (RGIDKRHWNSQ) derived from brain-derived neurotrophic factor (BDNF) to form RAD/RGI hydrogels. The RAD/RGI hydrogels were filled into the lumen of hollow chitosan tubes (HCST), which were then implanted into the SD rats of the sciatic neurotomies to bridge the nerve stump. The scaffold showed effects on the rats to promote axonal regeneration and increase the number of myelinated Schwann cells, showing a greater potential than unmodified RADA16-I to improve the treatment of sciatic nerve transection injury (Figure 4B) [40]. Another recent study [60] also showed that RAD/RGI hydrogels could promote the proliferation and neuronal differentiation of umbilical cord mesenchymal stem cells (UCMSCs) *in vitro*, and its co-transplantation with UCMSCs can promote the recovery of neurological function in rats with cerebral hemorrhage. Rat dorsal root ganglion (DRG) explants and rat DRG neurons were seeded on FGL-modified RADA16-I hydrogels (RAD/FGL). Using RADA16-I as a control, DRG explants on RAD/FGL were observed to have longer axons, and RAD/FGL promoted dorsal root ganglion neuron adhesion and axonal sprouting [61]. PFSSTKT is an active motif derived from bone marrow homing peptide 2 (BMHP2). The potential of PFSSTKT-modified RADA16-I (RAD/PFS) scaffolds for spinal cord nerve regeneration was evaluated in order to achieve regeneration of spinal cord injuries. The scaffold was implanted at the site of acute spinal cord injury in rats, and its early and late effects on injured tissue and motor function recovery were evaluated. The data showed that the scaffold promoted neural tissue growth, induced favorable matrix remodeling processes, and provided physical and nutritional support for tissue regeneration. Finally, the scaffold increased cell infiltration and axonal regeneration/sprouting, improving motor recovery in rats [62].

Modification of RADA16-I with related active motifs can effectively enhance the expression of nucleus pulposus cells (NPCs) and promote NPCs regeneration. The active motifs derived from bone morphogenetic protein 7 (BMP7) were widely studied in this direction. Three active motifs derived from BMP7 (SNVILKKYRN and KAISVLYFDDS) were used to modify RADA16-I, and the effects of these functionalized RADA16-I on human degenerated SNPCs were investigated. The results showed that both functionalized RADA16-I showed good biocompatibility, strong adhesion to cells, and promoted cell migration. The cell proliferation rate and levels of collagen II and aggrecan secretion of cells after being cultured for 28 days in the RAD/SNV scaffold were significantly higher than those in RADA16-I or RAD/KAI scaffolds, which indicates the great potential of RAD/SNV scaffold in nucleus pulposus tissue engineering applications [63]. Other recent studies [64] have independently investigated the effect of SNVILKKYRN-modified RADA16-I hydrogels on adipose tissue-derived stem cells and found that the hydrogels promoted the differentiation of stem cells into spinal nucleus pulposus-like cells. RLN (DHLSDNYTLDHDRAIH) is the amino-terminal peptide of connexin and one of the degradation products of connexin. A study on the effect of RLN-modified RADA16-I scaffolds on rabbit NPCs showed that the scaffold promoted cell adhesion and migration and significantly stimulated cell biosynthesis of extracellular matrix (proteoglycan and type II collagen) [65].

**Figure 4 polymers-14-05221-f004:**
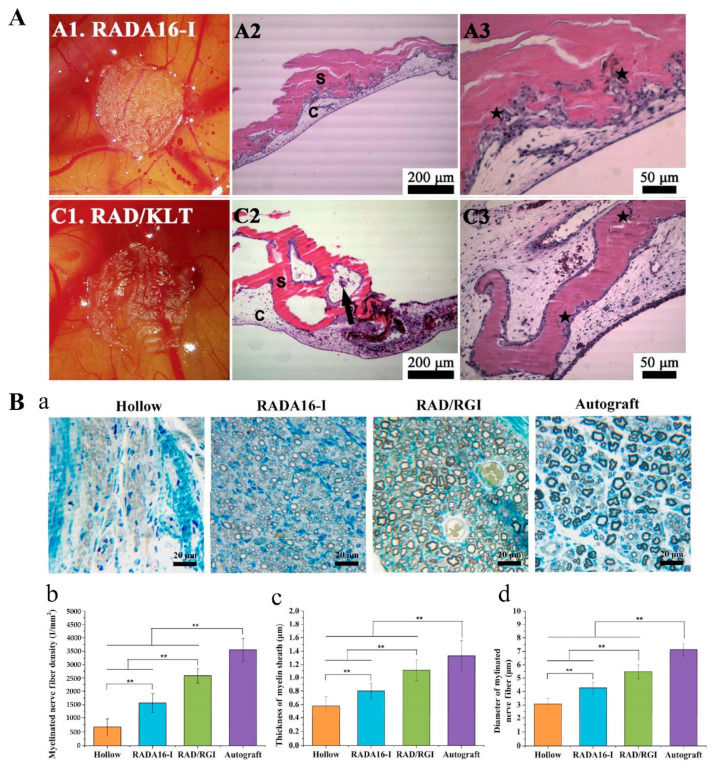
Promoting positive factors. (**A**) Gross morphology and H&E staining of CAM with different scaffold hydrogels at day four after implantation. S: implanted hydrogels; C: the CAM. Black arrows indicate new capillary vessels. ★ indicates that epithelial cells from the ectoderm of CAM proliferate and migrate into the implanted hydrogels. Adapted with permission from Ref. [57]. Copyright 2012 Nanoscale. (**B**) Morphometric analysis of the middle of the implants 12 weeks after surgery. Hollow: hollow chitosan nerve conduit group. Autograft: autologous nerve graft-transplanted group. (**a**) Toluidine blue staining of semi-thin sections. (**b**) The density of myelinated axons is expressed as the number of myelinated axons per square millimeter (n, 1/mm^2^). (**c**) The diameters of myelinated axons. (**d**) The thickness of myelin sheath. ** *p* < 0.01 (*n* = 3). Adapted with permission from Ref. [40]. Copyright 2018 Nano Research.

##### Resisting Negative Factors

RADA16-I can be modified with active motifs to counteract harmful factors detrimental to cell growth, tissue defect repair, or disease treatment. Cancer therapy is to eradicate or significantly block the growth of primary tumors and is thus most relevant to this strategy.

Melittin (GIGAVLKVLTGLPALISWIKRKRQQ) is a bee venom-derived cationic polypeptide consisting of 26 amino acids with great anticancer potential. A study [42] showed a significantly higher tumor cell growth inhibition rate with melittin-RADA32-doxorubicin hydrogels (MRD) than that with the RADA16-I group mixed with doxorubicin (RD) (Figure 5A). In addition, flow cytometry analysis showed that the percentage of necrotic cells in the MRD hydrogel group reached 40.5%, which was 5.1 times higher than that in the RD hydrogel group (Figure 5B). Subsequent experiments also showed that MRD hydrogels inhibited lymphatic tumor metastasis better than RD hydrogels and had immune activation and long-term immune memory effects.

The acidic environment can inhibit the proliferation, differentiation, and ECM production of human nucleus pulposus mesenchymal stem cells (hNPMSCs) through acid-sensing ion channels (ASICs) and thus affect the biological activity of hNPMSCs for the treatment of intervertebral disc degeneration. Therefore, if ASICs can be blocked, the adverse effects of the acidic environment in the intervertebral disc (IVD) on hNPMSCs can be alleviated, and the regeneration of the nucleus pulposus can be promoted. EDVDHVFLRF(Sa12b), a peptide related to FMRFa, extracted from the venom of the solitary bee silver-spotted butterfly, can be used as a blocker of acid ion channels. Sa12b was conjugated to the C-terminus of RADA16-I to obtain RSA1 hydrogels. The effect of RSA1 on hNPMSCs was assessed in an acidic environment. The results showed that RSA1 promoted the proliferation of hNPMSCs in an acidic environment and up-regulated the expression of collagen type II, SOX-9, and aggrecan (Figure 5C) [66].

GSK-3β and calpain are thought to be involved in aberrant phosphorylation of tau protein and may play an important role in neurodegenerative diseases. Treatment with GSK-3β-specific inhibitors significantly reduced the phosphorylation of Tau proteins and prevented Tau-induced neurodegeneration in mice [67]. RADA16-I was modified with a neural cell adhesion molecule-derived (NCAM) mimetic peptide (SIDRVEPYSSTAQ) and loaded LiCl onto functionalized RADA16-I to form a drug-loaded hydrogel scaffold. The scaffold can reduce the phosphorylation of Tau by inhibiting the calpain/GSK-3β signaling pathway, thereby significantly promoting cell proliferation, inhibiting apoptosis, and significantly increasing axonal length, number, and branch number [68].

KPSSAPTQLN (KPS) is an active motif derived from BMP-7 and can attenuate the harmful effects induced by tumor necrosis factor-alpha (TNF-α) by inhibiting NF-kB signaling. RADA16-I was modified with KPS (RADA-KPSS) and showed that this functionalized RADA16-I could attenuate inflammatory mediators, decrease matrix-degrading enzyme gene expression, reduce NPC apoptosis, enhance ECM protein expression, and inhibit NF-κB signaling in TNF-α-treated NPCs [69]. Another active motif-modified RADA16-I, LDWSWL- RADA16-I can also block the TNF-α signaling pathway to affect osteoblast differentiation [70].

### 2.4. Mixed Use of Biologically Active Motifs

Injury repair and disease recovery are often multifactorial and multiprocessing. Functionalizing RADA16 with a single motif may be difficult to address complex therapeutic needs. By modifying RADA16 with active motifs with active synergistic or complementary properties and using them in combination, it is possible to consider multiple factors and processes, thereby improving the clinical effect of RADA16 as a biomaterial.

RADA16-DGDRGDS and RADA16-IKVAV motifs to investigate their repair effects on cerebral hemorrhage, where the motif DDGRGDS can promote nerve cell adhesion, the IKVAV can promote and guide neurite outgrowth, and the two active motifs have opposite charges under physiological pH conditions. Mixing the two motifs has a synergistic effect both biologically and electrically. Injecting the mixed functionalized RADA16-I into the site of intracerebral hemorrhage in mice was found to reduce the number of apoptotic cells, glial responses, and inflammatory responses, thereby reducing acute brain injury and promoting functional recovery [71]. The mixture of two functionalized RADA16-I loaded with various active factors was used to treat spinal cord injury with the sustained release effect of active factors with different electrical properties in RADA16-DDGRGDS (-) or RADA16-IKVAV (+) being investigated. It was found that the active factor with the opposite electrical property to the active motif had a longer sustained release than the active factor with the same electrical property as the active motif. RADA16-DGDRGDS and RADA16-IKVAV were loaded with oppositely charged active factors, then mixed. The active factor-loaded mixed hydrogels greatly promoted the proliferation of endogenous neural stem cells (eNSCs), differentiation into neurons, maturation, and myelination [72].

Brain-derived neurotrophic factor (BDNF) is neuroprotective and can promote the repair and regeneration of damaged neurons, while nerve growth factor (NGF) plays a key role in the survival, differentiation, and growth of neurons. RADA16-I was modified with CTDIKGKCTGACDGKQC and RGIDKRHWNSQ, derived from NGF and BDNF, respectively, and the two modified RADA16-I were mixed to prepare hydrogels. The mixed hydrogels can significantly promote neurite outgrowth of PC12 cells and functional recovery in mice with sciatic nerve defects [73].

Both repair of damaged neurons and angiogenesis, which provide oxygen and nutrients, are required in nerve injury. VEGF can promote the proliferation and migration of vascular endothelial cells, leading to angiogenesis. BDNF can promote neuronal survival and differentiation. RADA16-I was modified with the VEGF-derived KLTWQELYQLKYKGI and BDNF- derived RGIDKRHWNSQ, respectively, and the two modified RADA16-I were mixed to prepare hydrogels [74]. In vitro cell experiments showed that the mixed hydrogels scaffold effectively promoted Schwann cell promyelination, as well as endothelial cell adhesion and proliferation. The mixed hydrogels significantly increased the number of new blood vessels and the density of regenerated axons in the rat sciatic nerve defect model. A synergistic effect of the combination of the two modified RADA16-I was proved by the better performance of the mixed hydrogels than any one of the two. Another study [75] also showed that the combination could promote spinal cord regeneration by directing regenerating tissue, accelerating axonal regeneration, sheath formation, and promoting angiogenesis.

## 3. Modification of RADA16-I for Mechanical Performance

Modification of RADA16-I by active motifs endows RADA16-I with specific biological activity, but modification in activity alone is not sufficient. The weak mechanical strength of RADA16-I hydrogels structure constructed by the weak interaction force of non-covalent bonds brings great challenges for applications in some tissues, such as large bone defect regeneration and ligament repair, where the repair materials are required not only for a certain biological activity but also a strong mechanical performance as support. Therefore, there is a great need for RADA16-I to enhance its mechanical performance and adjust its degradation rate adjusted.

The most common and easiest way to enhance the mechanical performance of RADA16-I is to form a composite material by mixing RADA16-I with materials with stronger mechanical performance. In this composite material, RADA16-I is responsible for providing suitable biological activity, while the mechanically stronger materials provide structural support. The silk sponge/fiber scaffold was primed with RADA16-I, and a silk-peptide hybrid scaffold system was constructed for ligament repair by this approach [76]. In this multilayer structure, the outermost layer of RADA16-I, similar to the extracellular matrix, played a role as a landing site for cells and promoted cell adhesion and proliferation. The silk sponge/fibers had strong mechanical properties and could resist a certain mechanical impact during ligament tissue repair. In vitro experiments showed that silk-peptide hybrid scaffolds can promote the proliferation, metabolism, and fibroblast differentiation of BMSCs as well as the secretion of total collagen and glycosaminoglycan from BMSCs within three weeks of culture. Poly (lactic-co-glycolic acid) (PLGA) was mixed with RADA16-I-BMHP1 solution in a certain proportion to obtain a PLGA-peptide scaffold. SEM imaging showed that the self-assembling structure of RADA16-I-BMHP1 can be observed on PLGA nanofibers that provide structural and mechanical support functions (Figure 6A). The effect of this scaffold on Schwann cells and the sciatic nerve transection rat model was evaluated, and the results indicated that the PLGA-peptide scaffold significantly promoted the genotype higher expression of markers and bipolar extension of Schwann cells, the regeneration of natural collagen tissue and myelin sheath and the recovery of sensorimotor and motor functions in mice model [77,78,79]. The RADA16/CaSO_4_/HA scaffold was prepared by filling the pore structure and surface of CaSO_4_/nano-hydroxyapatite (CaSO_4_/HA) material with RADA16-I [80]. The RADA16/CaSO_4_/HA scaffold loaded with bFGF increased the expression of typical markers of osteogenic differentiation of MC3T3 cells and repaired the femoral condyle defect in rats, which indicated the great potential of the scaffold to be applied to bone regeneration in large bone defects. Similarly, a mixture of nano-hydroxyapatite/polyamide 66 (nHA/PA66) and D-RADA16-RGD for repairing large bone defects had also achieved promising results [81].

In addition to the simple mixing mentioned above, filling RADA16-I in structures formed by chemical cross-linking other materials to form an interpenetrating network (IPN) is also an effective method. 1% RADA16-RGD was blended with 5% photocrosslinkable diacrylated poly(ε-caprolactone)-b-poly(ethylene glycol)-b-poly(ε-caprolactone) triblock copolymer (PCECDA), and then PCECDA was chemically cross-linked in the mixture by UV irradiation. Through this treatment, the non-covalent RADA16-RGD polypeptide and covalently cross-linked PCECDA formed an interpenetrating network of hydrogels (1% RADA16-RGD/5% PCECDA), with the storage modulus (G) being increased to about 2000 Pa, greater than the G’ of 1% RADA16-RGD peptide hydrogel (~700 Pa). When the concentration of PCECDA was increased to 8% (1% RADA16-RGD/8% PCECDA), the G’ of the hydrogel was even increased to ~4000 Pa (Figure 6D–F). By adjusting the concentration of PCECDA in IPN, the mechanical strength of the hydrogel can be easily adjusted. The hydrogels can significantly reduce cavitation, glial scar formation, and inflammation at the injury sites in the rat model with hemisection spinal cord injury [82]. Temperature-sensitive IPN hydrogels with mechanical and physical properties similar to brain ECM were prepared by mixing N-isopropylacrylamide HA conjugates with RADA16-I in different composition ratios [83]. Shigehito Osawa and his colleagues [84] blended three components, RADA16-I, Chitosan (CH), and *N*-hydroxysuccinimide ester-terminated poly-(ethylene glycol) (HS-PEG-NHS), to form a hydrogel scaffold (CH/PEG/RADA16), in which HS-PEG-NHS acted as a cross-linking agent to cross-link CH to form IPN. The elastic modulus of CH/PEG/RADA16 was higher than that of CH/PEG (the mixture of Chitosan and PEG) and RADA16-I hydrogels. The IPN hydrogels were encapsulated with human auricular chondrocytes and transplanted into the back of mice subcutaneously. It was observed that the construction of the IPN structure effectively promoted the formation of hyaline cartilage tissue by chondrocytes. To improve the degradation of the IPN hydrogels, Shigehito Osawa and colleagues then replaced HS-PEG-NHS with the more easily degradable poly(ethylene glycol)-block-poly(DL-lactide)-block-poly(ethylene glycol) (NHS-PEG-b-PLA-b-PEG-NHS). Chondrocytes cultured in the Modified IPN hydrogels exhibited higher levels of protein secretion and gene expression than the original combination [85].

Another approach to enhance mechanical properties was using short peptide motifs to modify RADA16-I. The new sequences R3 (n-RADARADARADARADA-GGAGGS-c) and R4 (n-RADARADARADARADA-GPGGY-c) by covalently binding spider silk GGAGS or GPGGY amorphous motifs to the C-terminus of RADA16-I can self-assemble into hydrogels, and the modification with both of the spider silk motifs effectively enhanced the mechanical strength of RADA16-I (Figure 6B,C) [86]. RADA16-I was modified with the QQLK motif for hemostasis in another study [87], The hydrogels coupled with the QQLK functional motif could be cross-linked under the action of endogenous transglutaminase secreted during coagulation and the mechanical strength of the cross-linked hydrogel was significantly increased.

**Figure 6 polymers-14-05221-f006:**
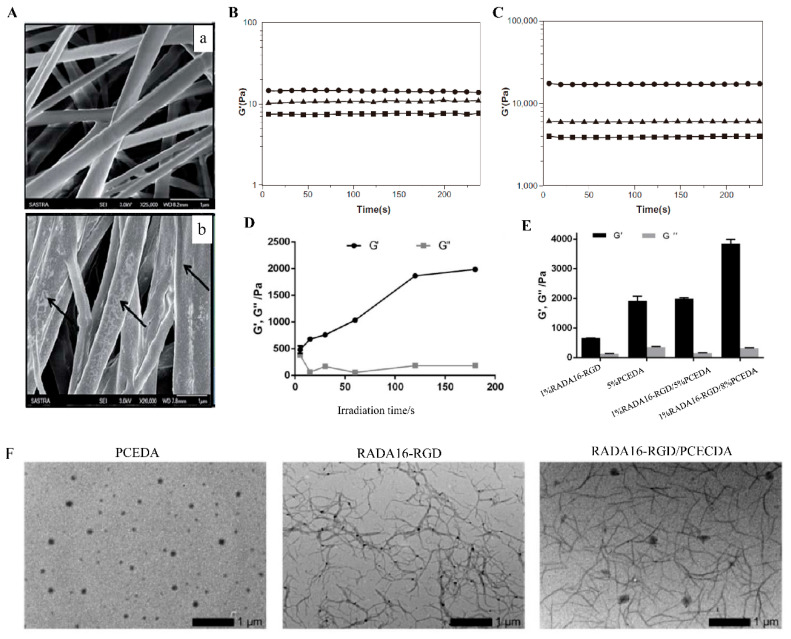
Modification of RADA16-Ifor mechanical performance. (**A**) SEM images of 200–300 nm PLGA nanofibers (**a**) and PLGA nanofibers coated with 1% peptide (**b**). Adapted with permission from Ref. [79]. Copyright 2015 RSC advances. (**B**) The storage modulus of RADA16-I, R3, and R4 in aqueous solutions. (**C**) The storage modulus of RADA16-I, R3, and R4 after they have formed hydrogels in the medium. Adapted with permission from Ref. [86]. Copyright 2012 International Journal of Nanomedicine. (**D**) G′ and G′′ of the 1%RADA16-RGD/5%PCECDA. (**E**) G′ and G′′ of 1%RADA16-RGD, 5%PCECDA, 1%RADA16-RGD/5%PCECDA and 1%RADA16-RGD/8%PCECDA hydrogels, respectively. (**F**) TEM characterization of the RADA16-RGD (1% *w*/*v*), 5% PCECDA (5% *w*/*v*) and RADA16-RGD/PCECDA (6% *w*/*v*, mRADA16-RGD/mPCECDA = 1:5) hydrogels, respectively. Adapted with permission from Ref. [82]. Copyright 2020 Biomedical Materials.

## 4. Modification of RADA16-I for Drug Delivery

RADA16-I can be used for the delivery of small molecule drugs [18,88,89], various active factors [90,91], or cells [92]. The release of drugs or active factors in RADA16-I hydrogels can be affected by various factors. On the one hand, various properties of the loaded drugs or active factors, such as hydrophilicity, hydrophobicity, molecular size, type and number of charges, etc., [30,93,94] can affect the release. On the other hand, for a specific drug or active factor, controlled release or sustained release can only be achieved through the modification of RADA16-I.

Appropriate changes to the original sequence of RADA16-I can make it suitable for the delivery of specific drugs or active factors. The sixth amino acid of RADA16-I was replaced with phenylalanine with an aromatic ring side chain (to obtain RADA16-F6) for packaging 5-fluorouracil (5-FU) since 5-FU contains an aromatic pyrimidine ring that can interact effectively with the phenylalanine side chain in RADA16-F6 in a π-π stacking interaction. The experiment results showed that RADA16-F6 could form hydrogels under neutral pH conditions and achieved a higher encapsulation rate for 5-FU than RADA16-I, which indicated that RADA-F6 hydrogel could be used as a pH-responsive drug delivery system for 5-FU (Figure 7A) [95]. In another study [96], in order to improve the sustained release effects of RADA16-I on hydrophobic drugs, multiple hydrophilic amino acids in RADA16-I were replaced with less hydrophilic glycine. The resulting sequence, Ac-RADAGAGARADAGAGA-NH2, can stabilize the hydrophobic model compound drug pyrene in water and encapsulate it to form a colloidal suspension.

RADA16-I can also be covalently bound to synthetic polymer materials to improve the delivery of drugs or active factors. Li et al. [97] used Maleimide-polyethylene glycol (PEG) -PLGA to modify RADA16-I to form a drug depot (RDDs), followed by VEGF being loaded into RDDs (V-RDDs) by simply mixing RDDs with VEGF and the V-RDDs being added to RADA16-I hydrogels. In the design, PLGA in RDDs was used to encapsulate VEGF, and PEG was used as a spacer to expose the modified RADA16-I, while the exposed modified RADA16-I could participate in the self-assembling of unmodified RADA16-I, thus forming a complex VEGF sustained-release delivery system. The release kinetic studies in vitro confirmed that V-RDDs-anchored hydrogels have desirable properties for long-term sustained release of VEGF (Figure 7C,D). In vivo experiments showed that long-term local release of VEGF from V-RDDs significantly enhanced EPCs-induced neovascularization. The research team also expanded the RDDs slow-release system to the delivery of tacrolimus, and the reduced immune rejection by slow-release tacrolimus and improved stem cell survival indicated the applicability of the slow-release system for delivering a variety of drugs [98].

RADA16-I can be modified with a sequence that has a stronger affinity to certain drugs or active factors to achieve specific sustained-release effects. RADA16-I was modified with the heparin-binding domain LRKKLGKA, the formed RAD/LRK hydrogels can bind to heparin through the heparin-binding domain, and the heparin bound in the hydrogels can then bind a variety of heparin-binding growth factors [99]. In this study, RAD/LRK was mixed with heparin for the delivery of VEGF and injected into the rat myocardium. Subsequent studies showed that the RAD/LRK hydrogel was effective in providing sustained delivery of VEGF for at least one month. In contrast, when VEGF alone or VEGF plus RADA16-I was injected, 75% or 87.5% of VEGF was released after one day, and almost no growth factor molecules could be detected at the injection site at 28 days. In vivo experiments demonstrated that RAD/LRK and heparin hydrogels were more beneficial to the induction of angiogenesis through sustained release of VEGF than the control group (Figure 7B). In another study, RAD/LRK sequences loaded with VEGF and HGF were also applied to promote angiogenesis [87].

**Figure 7 polymers-14-05221-f007:**
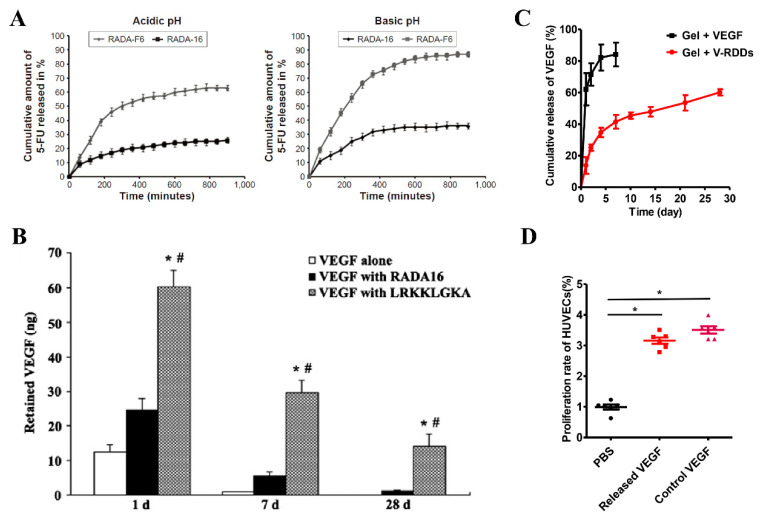
Modification of RADA16-Ifor drug delivery. (**A**) In-vitro 5-FU release profile from RADA-F6 and RADA-16 at two different pH, namely, acidic (pH 2) and basic (pH 7.8). Adapted with permission from Ref. [95]. Copyright 2016 International journal of nanomedicine. (**B**) Sustained release of VEGF by RAD/LRK (LRKKLGKA) over 28 days. * *p* < 0.01 versus the RADA16 group. ^#^
*p* < 0.01 versus the VEGF group. Adapted with permission from Ref. [99]. Copyright 2012 Biochemical and biophysical research communications. (**C**) Release profiles of VEGF from the RADA16 hydrogel alone (■) or the V-RDDs-anchoring-hydrogel (●) over 28 days (*n* = 3); (**D**) Bioactivity of VEGF released from the V-RDDs-anchoring hydrogel comparing with those of PBS (negative control) or free VEGF (positive control), (*n* = 6, * *p* < 0.05). Adapted with permission from Ref. [97]. Copyright 2017 Journal of Controlled Release.

**Table 2 polymers-14-05221-t002:** List of the source, application fields, and partial activity of modified RADA16-1 active motifs.

Active Motifs	Source	Fields of Applications or Purpose	Partial Activity	References
RGDS	Fibronectin and laminin	Nerve and skin regeneration	Promoting cells adhesion	[37,100]
PRGDSGYRGDS	Collagen VI	Nerve and bone regeneration, angiogenesis	Promoting cells adhesion, proliferation and migration	[35,37,51,52,57,58,101]
IKVAV	Laminin	Nerve and nucleus pulposus regeneration	Stimulated differentiation of cells into neurons, reduced pro-inflammatory cytokines	[37,41,53]
YIGSR, SDPGYIGSR	Laminin	Nerve and periodontal ligament regeneration	Promoting cells adhesion, spreading and neuronal differentiation	[37,52,54]
PFSSTKT	BMHP2	Nerve and spinal cord regeneration	Promoting cells spreading and axon regeneration	[37,62]
SKPPGTSS	BMHP1	Nerve and cartilage regeneration	Promoting cells proliferation and recruitment	[37,101,102]
SVVYGLR	Osteopontin	Angiogenesis, brain injury repair	Low biological activity	[33,35]
c(RGDfK)	RGD derived motif	Nerve regeneration	Cell migration and neurite elongation	[50]
PDSGR	Laminin	Nerve regeneration	Low biological activity	[37]
FLGFPT	Myelo-peptides family	Nerve regeneration	Cells spreading	[37]
EVYVAENQGGKSKA	NCAM	Nerve regeneration	promoting neurite outgrowth	[61]
RGIDKRHWNSQ	BDNF	Nerve regeneration	Promoting the proliferation and neuronal differentiation of UCMSCs	[40,60]
SIDRVEPYSSTAQ	NCAM	Nerve regeneration	Promoting neuronal attachment, proliferation and axonal extension	[68]
KLTWQELYQLKYKGI	VEGF	Angiogenesis	Enhanced endothelial cells proliferation, migration, and morphological tubulogenesis	[35,57,58]
VGVAPG	Elastin	Angiogenesis	Low biological activity	[35]
REDV	Fibronectin	Angiogenesis	Low biological activity	[35]
LKKTETQ	Thymosin β4	Angiogenesis	Low biological activity	[35]
SNVILKKYRN	BMP7	Spinal nucleus regeneration	Promoting stem cells differentiation into NP-like cells and ECM biosynthesis	[63,64]
KPSSAPTQLN	BMP7	Spinal nucleus regeneration	Stimulating ECM biosynthesis, inhibition of NF-κB signaling	[103,104]
DHLSDNYTLDHDRAIH	Link protein	Spinal nucleus regeneration	Promoting cells adhesion, migration and ECM biosynthesis	[65]
EDVDHVFLRF	The venom of butterfly	Spinal nucleus regeneration	Inhibition of p-ERK expression	[66]
KAISVLYFDDS	BMP7	Spinal nucleus regeneration	Low biological activity	[63]
RLNSDNYTLDHDRAIH	Link protein	Spinal cord, cartilage regeneration	Promoting cells adhesion and ECM biosynthesis	[105]
RGDSP	Fibronectin	Stem cells transplantation	Enhanced survival and differentiation of cardiac stem cells	[106]
FPGERGVEGPGP	Type I collagen	Skin regeneration	Promoting cell migration, inhibiting cells proliferation	[37,100]
DGRGDSVAYG	Osteopontin	Bone regeneration	Enhanced mouse preosteoblast cells proliferation and differentiation	[51]
FHRRIKA	Heparin-binding motif	Stem cell culture	Maintaining the growth factor secretion capacity of stem cells	[101]
ALKRQGRTLYGF	Osteogenic Growth Peptide	Bone regeneration	Enhanced mouse preosteoblast cells proliferation and differentiation	[51]
LDWSWL	The IkB kinase complex	Bone regeneration	Promoting the osteogenic differentiation of MC3T3 E1 cells	[70]
CDDYYYGFGCNKFCRPR	The Notch ligand Jagged-1	Myocardial infarction	Stem cells recruitment and angiogenesis	[107]
QHREDGS	Angiopoietin	Myocardial infarction	Promoting angiogenesis and paracrine	[108]
FRKKWNKWALSR	Proadrenomedullin	Induce mast cell activation	Activation of mast cells	[39]
GIGAVLKVLTGLPALISWIKRKRQQ	Bee venom	Anti-cancer	Effective in vivo antitumor	[42]
IPQVS, ELHQEEPL	Rapeseed napin	Diabetes treatment	Inhibiting dipeptidyl peptidase-IV and increasing Glucagon-like peptide-1 release	[109]
GGAGGS	Spider silk	Improving mechanical performance	The storage modulus of RADA16-I is increased by 0.5 times	[86]
GPGGY	Spider silk	Improving mechanical performance	The storage modulus of RADA16-I is increased by 3 times	[86]
QQLK	A cross-linking peptide	Improving mechanical performance	Improved mechanical performance of RADA16	[87]
GSVLGYIQIR	A Ca^2+^ binding peptide	Improving mechanical performance	Self-assembly of functionalized RADA16 into a mesh network	[110]
LRKKLGKA	Heparin binding consensus sequence	Drug delivery	Slow release of VEGF and HGF for 28 days	[99]

## 5. Conclusions and Outlook

The lack of specific activity, the poor mechanical performance, and the difficulty in achieving slow and controlled release for some drugs or active factors limit the application of RADA16-I as a promising biomaterial, especially as a drug carrier. Its modification is a topic worth discussing. In order to address the lack of specific activity of RADA16-I for specific tissues or diseases, it is often modified with active motifs, which are mainly derived from the extracellular matrix or the active domains of some active factors. The strategies of modification with active motifs include: enhancing the existing functions of cells and directly promoting normal cell growth and disease recovery, resisting negative factors that are not conducive to normal cell growth and disease recovery. In order to improve the mechanical performance of RADA16-I, it can be directly mixed with certain materials through physical cross-linking, mixed with substances that can undergo chemical cross-linking to form an interpenetrating network structure, or modified with some short peptide motifs. In order to make RADA16-I more suitable for the delivery of specific drugs or active factors, its sequence can be suitably modified to make it more suitable for the delivery of a particular drug or active factor. It can also be modified with polymer materials acting as drug reservoirs or with molecules that have a stronger affinity for specific drugs or active factors. Although there are some solutions (adding glycine, mixing the modified peptide with RADA16-I, and prolonging the original sequence) to make up for the destruction of self-assembling after modification, it is still necessary to follow the principle that the modification of RADA16-I should not destroy or only destroy its self-assembling property to as less as possible extent.

Although only the modification strategy of RADA16-I was discussed here, the modification strategies can also be applied to other ion-complementary self-assembling peptides. Hopefully, the strategies reviewed in this paper can provide some references for further basic research and clinical application of ion-complementary self-assembling peptides and their derivatives.

## Figures and Tables

**Figure 2 polymers-14-05221-f002:**
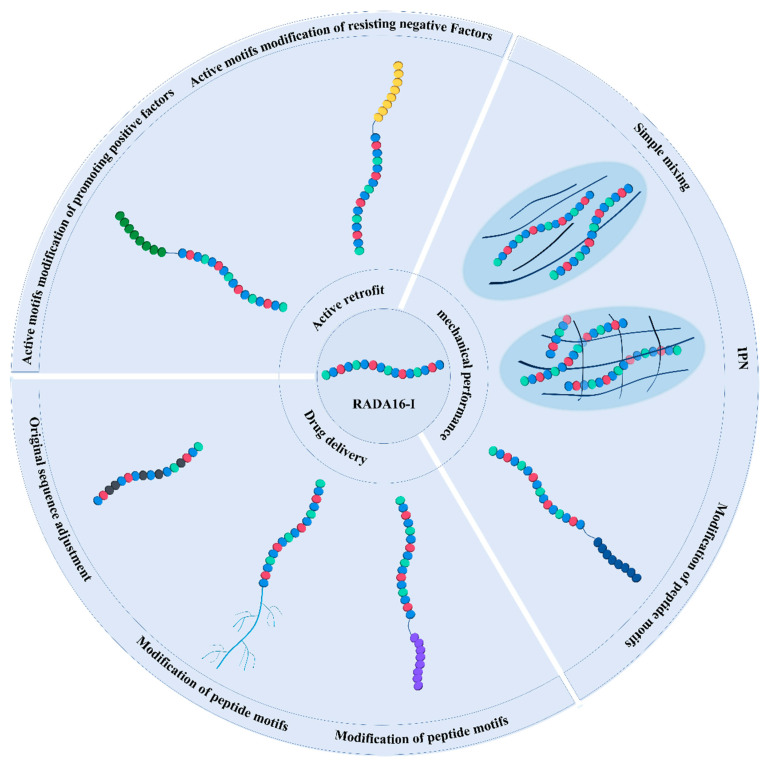
Overview of modification strategies for RADA16-I.

**Figure 5 polymers-14-05221-f005:**
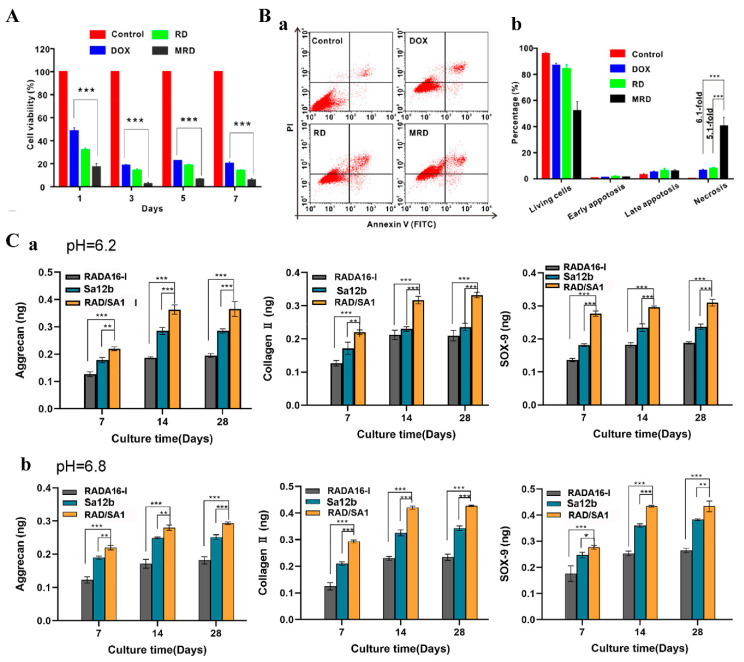
Resisting negative factors. (**A**) Measurement of cell viability in MRD hydrogel treatment using the CCK-8 assay. (**B**) Flow cytometry measurement of cell apoptosis induced by DOX, RD, or MRD hydrogel. Data are presented as the mean ± SEM (*n* = 3). Adapted with permission from Ref. [42]. Copyright 2018 Acs Nano. *** indicates *p* < 0.001. (**C**) ELISA revealed that the amounts of collagen II, aggrecan, and SOX-9 secreted by hNPMSCs were significantly different among the three groups (RAD/SA1, Sa12b, and RADA16-I) after 7, 14, and 28 days, respectively. (**a**) hNPMSCs were cultured at pH 6.2. (**b**) hNPMSCs were cultured at pH 6.8. When the two groups are compared, * indicates *p* < 0.1, ** indicates *p* < 0.01, and *** indicates *p* < 0.001. Adapted with permission from Ref. [66]. Copyright 2022 Frontiers in Cell and Developmental Biology.

**Table 1 polymers-14-05221-t001:** Some classical ionic complementary self-assembling peptides.

Name	Sequence	Charge Distribution	References
EAK16-II	AEAEAKAKAEAEAKAK	− − + + − − + +	[10]
RADA16-I	RADARADARADARADA	+ − + − + − + −	[11]
KLD12	KLDLKLDLKLDL	+ − + − + −	[12]
KFE8	FKFEFKFE	+ − + −	[13]

## Data Availability

Not applicable.

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
