# Peer review of "Modification Strategies for Ionic Complementary Self-Assembling Peptides: Taking RADA16-I as an Example"

_polymers, 2022, doi:10.3390/polym14235221_

Round 1

Reviewer 1 Report

The authors in this manuscript overviewed recent progresses on modification strategies for ionic complementary self-assembling peptide especially RADA-I. The modification of RADA-I for extracellular matrix, strong mechanical properties, and drug delivery are described. The manuscript is well-written for broad readers and the perspective describes the important subjects to be achieved in this area. Therefore, I think this review article would be suitable for the publication on Polymers after minor modification for a following comment.

1)      The authors should show each sequence of ion-complementary self-assembling peptides: EAK16-II, RADA16-I, and KLD12.

2)      The authors should cite the first original paper for self-assembly of RADA-I to form hydrogel. Probably the first paper is H. Yokoi, T. Kinoshita, S. Zhang, PNAS, 102, 8414-8419 (2005) DOI: 10.1073/pnas.0407843102.

Reviewer 2 Report

The manuscript entitled <Modification strategies for ionic complementary self-2 assembling peptides: taking RADA16-Ⅰ as an example> is a very well designed review article on quite highlighted subject which may attract a lot of Polymers readers. The topic of the manuscript fits well the scope of the Polymers Journal. This review clearly summarize the state-of the-art in the specific field of the research which is clearly substantiated by the relevant references. The conclusions and outlook sounds very well. The technical quality of the manuscript is high. I expect that this manuscript will have definite impact on the specific field of the research and will gain many citations in future. I do not have any specific comments and I can recommend this manuscript for publication as it is.

Author Response

Dear Reviewer: Thank you for your professional and kind comments on our manuscript entitled “Modification strategies for ionic complementary self-assembling peptides: taking RADA16-Ⅰ as an example” (ID: Polymers-1960209). Your kind comments and recognition of our research will inspire us to do better. Kind regards Fushan Tang

Reviewer 3 Report

The manuscript for Polymers MPDI is a review about the possible modification of RADA16-I with different motives. The altered sequences are reported as tools for different application areas, including tissue engineering and drug delivery. 

The topic is interesting. I suggest, however, a major revision step before pubblication according to the following points:

1) It is important delineate the general scenario of peptide based hydrogels. I suggest to include and reference short and ultrashort sequences, multicomponent matrices and all-aromatic peptide sequences (e.g. doi.org/10.1039/D0SM00825G) 

2) It could be interesting report analogue sequences of RADA16-I. The peptide identification is missing.

3) The general aggregation properties of RADA16-I is totally missing (diffraction, miscroscopy, rheology, CD, FT-IR). 

4) Figure are totally missing. I suggest to include at least three additional figures. 

5) Chemical structures of peptide are missing. Include the more important ones. 

6) The general discussion about the modified peptide is poor (e.g. 2.2.1 and consectives) with respect to other ones. Please, enlarge all the paragraph with consistent data. 

Reviewer 4 Report

This review concerns the ionic complementary self-assembling peptides RADA peptide.

It is very interesting review, but should describe why this sequence was selected among the many peptides.

The results of a typical application should be explained with figures.

For examples where peptides are mixed, a description of mechanical strength should also be added.

Round 2

Reviewer 3 Report

I would like to thank the authors for the effort in the revision step. In my opinion the manuscript is now suitable for publication in present form.